# The Influence of Fly Ash on the Tensile Creep Prediction of High-Strength Concrete at Early Ages

**DOI:** 10.3390/ma16041337

**Published:** 2023-02-04

**Authors:** Jikai Yao, Shuifeng Yao, Senle Huang, Tongyuan Ni, Chenhui Jiang, Yang Yang, Deyu Kong

**Affiliations:** 1College of Civil Engineering and Architecture, Zhejiang University of Technology, Hangzhou 310023, China; 2Key Laboratory of Civil Engineering Structures & Disaster Prevention and Mitigation Technology of Zhejiang Province, Zhejiang University of Technology, Hangzhou 310023, China; 3Department of Construction Engineering, Zhejiang College of Construction, Hangzhou 311231, China; 4Zhejiang Huawei Building Materials Group, Co., Ltd., Hangzhou 310018, China

**Keywords:** fly ash, high strength concrete (HSC), prediction model, tensile creep, paste material factor

## Abstract

In this study, the tensile creep (TC) of high-strength concrete (HSC) containing 30 wt% fly ash was measured at early ages to investigate the applicability of creep prediction models for concrete containing FA, and to provide ideas to study the prediction model of concrete creep containing other SCMs in the future. The TC values obtained from the experiment were compared with the predicted values of six TC models. Then the accuracy of different models was evaluated by the ratio of predicted values to experimental values. Finally, the applicability of these models to the TC of HSC with fly ash was discussed at an early age. By comparison, it was found that when the loading age was 1d, 2d, and 3d, the ZC model (ZC are the initials for the word “Self-developed” in Chinese), which is a rheology-based model for TC, proposed by Yang.Y et al. agreed with the experimental values. The predicted values of the other five models deviated significantly from the tested ones. When the loading age was 5d and 7d, the calculated results of the ACI 2009R model were more accurate. Compared with the other five models, the time dependency of the paste with fly ash was considered in the ZC model, and parameter ***q*** of the ZC model was introduced in order to characterize the influence of fly ash on the paste at early ages. Therefore, this paper demonstrated both theoretically and experimentally that the ZC model can better predict the early-age TC of HSC with fly ash.

## 1. Introduction

It is well known that tensile creep (TC) is an important component of the volume change of high-strength concrete (HSC) at an early age [1,2]. The property of TC significantly affects the crack resistance of HSC, and it would have a heavy impact on a concrete structure’s durability [3,4]; the detection of cracks and early crack resistance research is very important [5]. The cement paste plays an important role in the study of the creep properties of concrete [6]. To improve the crack resistance and durability of HSC, it is common to improve the paste performance by adding some activated powders, such as silica fume (SF), blast furnace slag (BFS), fly ash (FA), and so forth [7,8,9]. These activated powders are called supplementary cementitious materials (SCMs). So, the paste’s material factor is an important influencing factor to TC at early ages, especially in HSC containing FA [10,11,12]. Moreover, it is a difficult task to build a comprehensive model to predict the early-age TC, although it is meaningful to guide engineering practices [13,14,15].

Although the adopted definition of creep implies a distinction from elastic deformation, the two phenomena of elastic and plastic deformation cannot be truly separated, according to previous studies [4,16]. Considering the deformation of any material under load, rheology consists of three fundamental types of deformation: elastic, plastic, and viscous. These types can appear as elastic-plastic or visco-elastic [14,17]. The idealized deformations which are used to build up real behavior are elastic, viscous, or plastic and are represented by a spring, a dashpot, and a friction element, respectively [18]. The basic elements mentioned above can be built up into different rheological models. There are two basic models, the Kelvin (or Voigt) model (or body) and the Maxwell model (or body).

Normally, the elastic modulus of early-age concrete’s cement paste is much smaller than that of the aggregate [19]. At the elastic deformation stage, the deformation of the aggregate itself is almost negligible for the deformation of concrete with the same aggregate content [4]. In this case, the deformation of the paste is the main contributor to the deformation of concrete. So, the paste’s property is an important factor in early-age creep. According to the rheological theory assumption, the creep is generated by the viscous flow of cement paste and the seepage of adsorbed water [20,21]. Due to the creep of the paste being in proportion to the imposed stress, and the limitation of the paste flow by the aggregates, the main contributor to the creep is the cement paste [22]. With the hydration reaction, the paste properties change rapidly at early ages, whereas the aggregates, as inert materials, remain almost constant in their physical properties, such as modulus of elasticity. 

Hence, it is necessary to consider the influence of paste performance when building the TC prediction model for early-age concrete. The FA has a significant effect on the hydration of paste, which directly affects the creep of concrete [12,23]. To explore the law of creep of HSC [9], it is important to study the properties of the paste containing FA at an early age. The TC model, which considers the influence of the paste’s property at an early age, is closer to the objective rules.

In this paper, the differences in the theoretical basis and the factors considered in six different tensile creep models were discussed. The HSC with 30 wt% FA of early-age tensile creep was experimentally obtained, and the tested values were compared with the previous TC prediction models. Additionally, the accuracy of different prediction models was calculated by the ratio of prediction values to measured values. Finally, the applicability of six early-age TC models of HSC with FA was analyzed and discussed.

## 2. Prediction Models of Early-Age TC Considering the Paste Property

The prediction models of TC were developed based on theoretical analysis, such as considering the ageing diffusion process, thermal activation process, microcrack, and other physical processes. The parameters would be determined from the regression according to the experimental data. The parameters were given the actual physical significance, such as loading age, relative environmental humidity, aggregates, concrete strength, hydration process, etc. [13,24,25,26,27,28]. Based on phenomenology, the current representative creep mechanisms mainly include mechanical deformation theory [4], plastic flow theory [26], viscous flow theory, and micro prestress-consolidation theory. Based on the extensive experimental results and theoretical analysis, some prediction models of concrete creep have been proposed, mainly including BP-2, B-3 [14], CEB-FIP [29,30], ACI209R [31], GL2000 [20,32], and ZC [8] models. 

Among them, the ZC model considered the factor that the paste materials were hardening at an early age [8,33,34,35,36]. The logical composition schematic of the ZC model is shown in Figure 1. In the ZC model, the Maxwell model was composed of Spring A and Dashpot A to reflect the elastic deformation and unrecoverable viscous deformation in concrete. The viscoelastic strain in concrete is represented by the parallel composition of the Kelvin and Hook model [8,33,36]. Considering the development of the paste during the early ages, the Spring C with the time-dependent modulus of elasticity was added to the Kelvin model, which was parallel to Dashpot B [4]. 

According to the ZC model, the creep compliance, specific creep, and creep coefficient can be expressed by Equations (1)–(3), respectively.
(1)$$J(t,t_{0})=\frac{1}{E_{t}(t_{0})}+\frac{\left\{1-\exp[-\varphi (t-t_{0})]\right\}}{\chi \varphi }+\frac{1}{E_{V}+E_{H}}\{1-\exp[-(1+\frac{E_{H}}{E_{V}})\frac{{\left(t-t_{0}\right)}^{1-r}}{q(1-r)}]\}$$
(2)$$C(t,t_{0})=\frac{\left\{1-\exp[-\varphi (t-t_{0})]\right\}}{\chi \varphi }+\frac{1}{E_{V}+E_{H}}\{1-\exp[-(1+\frac{E_{H}}{E_{V}})\frac{{\left(t-t_{0}\right)}^{1-r}}{q(1-r)}]\}$$
(3)$$\phi (t,t_{0})=\frac{E_{t}(t_{0})}{\chi \varphi }\{1-\exp[-\varphi (t-t_{0})]\}+\frac{E_{t}(t_{0})}{E_{V}+E_{H}}\{1-\exp[-(1+\frac{E_{H}}{E_{V}})\frac{{\left(t-t_{0}\right)}^{1-r}}{q(1-r)}]\}$$

From the above three functional equations, there are six parameters in the ZC model, which are 1/χφ, φ, $1/(E_{V}+E_{H})$, $E_{H}/E_{V}$, q and r. The parameters q and r are related to the properties of the paste, and the parameters 1/χφ, φ, $1/(E_{V}+E_{H})$, $E_{H}/E_{V}$ mainly reflect the influence of the loading age. The physical meaning of parameter 1/χφ represents the final specific tensile creep of the Maxwell body (it can be represented by $C_{M}(\infty ,t_{0})$), and the physical meaning of the parameter $1/(E_{V}+E_{H})$ in the ZC model is the final specific tensile creep of the Voigt and Hooker bodies (they can be represented by $C_{g}(\infty ,t_{0})$). The sum of both is the final specific tensile creep of the concrete. Considering the influence of the water-to-binder ratio and FA on the paste hardening process during early ages, two parameters (***r*** and ***q***) were introduced to the ZC model, which were quite significant in predicting the accuracy of the early-age concrete TC. It was found that the parameter ***r*** was stabilized at 0.4 for different loading ages and the FA contents in the previous study by Yang’s research team [7,24]. So, the effect of FA on creep is mainly reflected by parameter ***q***.

The creep prediction models were built based on different theoretical evidence or theoretical explanations and considering different factors. The factors considered in different creep models are summarized in Table 1. 

## 3. Raw Materials and Experimental Methods

### 3.1. Raw Materials and Mix Proportion

The Portland cement (P·O 42.5) used in this study met the Chinese standard GB175-2007, and the FA (class II) was in accordance with the Chinese standard GB/T1596-2017. The properties of the cement and FA are shown in Table 2, the particle size distribution of the powders is shown in Figure 2a, and the FA’s SEM (Equipment type: GeminiSEM 500, Zeiss, Jena, Germany) image is shown in Figure 2b.

The coarse aggregate of concrete was crushed stone, the size range was from 5 to 31.5 mm, the apparent density was 2700 kg/m^3^, and it met the requirements of pumping concrete according to the Chinese standard JGJ/T 10-2011 (Technical Specification for Concrete Pumping Construction). The river sand was used as fine aggregate; the fineness modulus was 2.40, and it had an apparent density of 2630 kg/m^3^. The poly-carboxylate super-plasticizer with a 30% water-reducing rate was used as water reducing agent.

The 0.3 water-binder ratio and the 30wt % FA are commonly used in engineering for HSC (Strength grade was C50) containing FA. Hence, the experimental results of HSC TC with a 0.3 water-to-binder ratio and 30wt % FA was selected to contrast the applicability of six different creep models in this study. The mix proportion of HSC containing FA is listed in Table 3. The concrete slump was controlled at 180 ± 20 mm.

### 3.2. Basic Experimental Parameters and Mechanical Property Index Stress Level Determination

#### 3.2.1. Stress Level Determination

At the early ages, when the stress level is too high, there are risks of the internal structure of the immature concrete being damaged. Still, when the stress level is too low, the various resistances during the loading process affect the experimental results. Based on the above, the stress level was selected as 0.3, i.e., the stress/strength ratio was 0.3.

#### 3.2.2. Loading Ages

The loading age strongly influences the development of concrete creep. In this study, the loading ages were set at 1d, 2d, 3d, 5d, and 7d, respectively. The basic parameters of the TC experiment and mechanical properties are shown in Table 4.

### 3.3. Tensile Creep Test Method

#### 3.3.1. Specimen Size

The same size of autogenous shrinkage and TC specimens were used in order to eliminate the experiment influences which come from the autogenous shrinkage and TC specimen size, and the illustrations are reported in [8,9,33].

#### 3.3.2. Curing Temperature Condition 

All specimens were cured in the same environment and illustrated in Refs. [8,9,33].

#### 3.3.3. Test Procedure

The test methods and procedures of TC and autogenous shrinkage were used as reported in Refs. [8,9,33]. As shown in Figure 3, the displacement transducer (CDP-10, TML) was installed at both ends of the autogenous shrinkage specimens. Additionally, before the first 1h of loading, the strain gauges (PL-90-11, TML) were adhered to test the length change of the specimen. A similar tensile creep test method was used by NI et al. [8,9,37], WEI [38], and SHEN [39,40].

Theoretically, the basic tensile creep strain (***ε_tensile creep_***) can be obtained from Equation 4:(4)$$\varepsilon_{\;tensile\;creep}=\varepsilon_{total}-\varepsilon_{elastic\;}-\varepsilon_{T}-\varepsilon_{as}$$

According to Figure 3, the basic creep strain can be calculated with Equation 5:(5)$$\varepsilon_{tensile\;creep}=\varepsilon_{B}-\varepsilon_{A}-\varepsilon_{elastic\;}$$

The parameters in Equation (5), i.e., ***ε _B_*** (=***ε _total_***), ***ε*** _A_ (=***ε*** _as_ + ***ε*** _T_), and ***ε _elastic_***, were obtained by experimental measurements.

## 4. Applicability Analysis of Different Six Creep Models

### 4.1. Calculation Parameters of Six Creep Models

The mechanisms of creep are complex. Some theoretical explanations for mechanisms of creep were proposed. The actual creep may involve two or more mechanisms. The creep prediction models with different parameters would have been developed because of the mechanisms’ focus differences and comprehensiveness insufficient. The calculated values of the parameters in the different models are shown in Table 5.

### 4.2. Calculation Parameters of Six Creep Models

Obviously, different creep models focus on different priorities based on different theoretical evidence. For example, some models (BP-2, B-3 models) predict dry creep and basic creep separately, while the MC2010 model, ACI209R model, and GL2000 model do not make a distinction. The difference between models militates against the comparison of model prediction results. Given this situation, the prediction results of different models were unified to the parameter of a specific creep ($C\left(t,t_{0}\right)$) for comparison in this study.

The results of the comparison of the specific creep of HSC are shown in Figure 4. When the tensile load was imposed at 1 d (Figure 4a), the predicted values of the ZC model were the most acceptable. Except for the ZC model, the predicted values of the other five models had large deviations from the experimental values. The B-3 model was relatively close to the experimental values among the five models except for the ZC model. However, it overpredicted the tensile creep of HSC within 5 d after loading, and the predicted values were small after 5 d. In general, the prediction values of the five models, except for the ZC model, were small. Especially for the ACI209R, GL2000, and MC2010 models, the experimental values were almost 15 times higher than their predicted values.

When the loading age was 2d (Figure 4b), the predicted values of the ZC model were still the closest to the experimental values. The B-3 model was overpredicted in the first 15d of loading, and the predicted value was close to the experimental value after 15d under loading. The predicted values for the other models were still smaller than the experimental values, but the gap between the predicted and experimental values was significantly smaller compared to the 1d loading.

The results of Figure 4c–e showed that the predicted values of the ZC model were still highly consistent with the experimental values after 3d loading. The predicted values of the B-3 model were much higher than the experimental values, and a similar pattern was observed for the BP-2 model. When the loading age was at 5d and 7d, the ACI209R model was closer to the experimental values, while the GL2000 and MC2010 models were still smaller than the experimental values.

### 4.3. Prediction Accuracy Analysis of Six Creep Models

To evaluate the accuracy of the creep model in predicting the tensile creep of HSC containing FA, the ratio of the predicted value to the experimental value was calculated. Generally, the closer the ratio is to 1.0, the higher the accuracy of the model [12]. 

In Figure 5, the results of the development of the ratio of six models are shown. When the tensile force was imposed at 1 d (Figure 5a), the errors of the ZC model exceeded the range of +20% at the beginning of the loading period. With the development of age, although the ratio fluctuated, the errors were still in the range of ±20%. The errors of the B-3 model entered the range of +20% after 2 d of loading, but there was a trend that the errors of the B-3 model exceeded the range of -20% at the late ages. The ratio of the other models differed significantly from 1.0, which indicated poor prediction accuracy.

As the results of Figure 5b indicate, the errors of the ZC model were in the range of + 20% when the tensile load was imposed at 1 d. After 5d, the deviation between the other models’ values and the experimental values was still large, but the ratios were closer to 1.0 than that of 1 d loading.

The accuracy of the models generally increased as the loading age was delayed. When the loading age was 3d (Figure 5c), the errors of the ZC model were in the range of + 20%. After 4 d of loading, the error of B-3 exceeded 20%, while the errors of the other models were significantly decreased and close to the range of ± 20%. When the loading age was 5d, and 7d (Figure 5d–e), the errors of the ZC model and GL2000 model were the range of±20%, while the errors of the B-3 model and BP-2 model were larger than + 20%. The error of the ACI2009R model was the smallest because the ratio was close to 1.0.

From the above analysis, it can be concluded that the ZC model was well-fitted to the experimental values when the loading age was 1d, 2d, and 3d. When the tensile force was imposed at 5 d and 7d, the errors of the ZC model and GL2000 model were about—20%, and the ratio of the ACI2009R model was closer to 1.0, i.e., the biases of the ACI2009R model were smaller. Generally, the ZC model is more accurate in predicting the early tensile creep of HSC containing FA.

## 5. Discussion

In recent years, Vandamme et al. [41] investigated the micromechanics of C-S-H by nanoindentation tests. The results showed that the C-S-H gel behaved as macro creep properties of concrete, which was consistent with the trend of the creep in concrete [42,43]. Previous studies have tended to focus on ordinary concrete, separate from the different paste properties in ordinary concrete [44,45,46]. Thus, the time-dependency of paste properties in HSC containing FA was more prominent. It was mainly reflected in the slower early hydration process and fewer hydration products. To accurately predict the early-age creep of HSC containing FA, it is necessary to build some parameters which reflect the development of paste hydration at an early age.

It can be seen from Table 1 and Table 5 that most of the existing creep models may pay less attention to the influence of paste property evolution on concrete creep at an early age. In the MC2010 model, the concrete tensile creep is predicted by the 28d compressive strength. This model may be more suitable for ordinary concrete but not for FA concrete. It is mainly because the influence of FA on the compressive strength at 28d is not consistent with its effect on the hydration of the paste at an early age. At an early age, the reaction activity of FA was low, and FA slowed down the hydration process of the paste. In contrast, the pozzolanic activity of FA improved at a late age, which accelerated the development of compressive strength. In the MC2010 model, the adjustment coefficient was not introduced to reflect the effect of FA on the properties of the paste.

The same problem existed in ACI209R, GL2000, and BP series models. The correlation coefficient considered in the ACI209R model is mainly related to loading age, environmental humidity, specimen size, fine aggregate content, etc. The Gl2000 model mainly considered the factors such as loading age, environmental humidity, and specimen surface ratio. In addition to these factors, the influence of FA on the early-age properties of paste was not reflected in these models. In the B-3 model, short-term creep and long-term creep were considered separately, which were adjusted by q2, q3, and q4 parameters. Due to the lack of correlation between these parameters and the influence of admixtures on paste properties, the prediction results were still biased. Generally, the BP-2, B-3, MC2010, ACI209R, and GL2000 models did not consider the variation of paste properties during hardening and the influence of FA on the properties of the paste at early ages, so they were not suitable for predicting the creep of HSC containing FA at an early age.

As shown in Figure 1, the Spring B and Kelvin models were used in the ZC model to represent the aggregate and paste in concrete, respectively. Spring C and Dashpot C in the Kelvin model represented the crystalline phase and gel term in the concrete, respectively. In the crystalline phase, the main components are CH crystals, calcium silicate hydrate, ettringite, and other hydration products, and the gelling item is mainly composed of C-S-H gel. To reflect the hardening process of the cement paste on the hydration process, two parameters (***r*** and ***q***) were introduced to the ZC model, and the stiffness of Spring C and the viscosity coefficient of Dashpot C were set to change with time, which reflected the process of concrete performance change at the early age. Because the structural composition and hardening process were considered in the ZC model, it is more accurate at reflecting the early-age tensile creep of HSC containing FA.

Although the ZC model showed good applicability in predicting early-age tensile creep of HSC with 30wt% FA, its applicability for other types of SCMs is still uncertain. The effect of different types of SCMs on paste properties is different. Therefore, the applicability of the prediction model to HSC containing other SCMs remains to be further investigated. Relevant experiments will be conducted in further study.

## 6. Conclusions

The early-age TC of HSC with 30wt % fly ash was measured in this study, and the experimental values were compared with the predicted values of six tensile creep models. The factors considered by different creep prediction models and the applicability of the models were discussed. The conclusions of this study can be summarized as follows:(1)The TC was significantly affected at early ages by the paste properties development and the FA. The rate of TC development was faster when the loading age was greater.(2)The comparison results between the tensile creep experimental values and the model predictions values of HSC with 30% FA of tensile creep and the model predictions showed that the errors of BP-2, B-3, MC2010, ACI 209R, and GL 2000 models were relatively larger than ZC model, especially when loading age was less than 3d.(3)The ZC model can be developed to consider the time-dependent property of a paste containing FA. The parameter ***q*** was introduced in order to reflect the influence of FA on paste, and it agreed well with the experimental values of early-age TC of HSC containing FA.(4)The effect of different types of SCMs on paste properties is different. The applicability of the TC prediction model to HSC containing other SCMs remains to be further investigated. A more comprehensive concrete creep prediction model could be useful for concrete production requirements.

## Figures and Tables

**Figure 1 materials-16-01337-f001:**
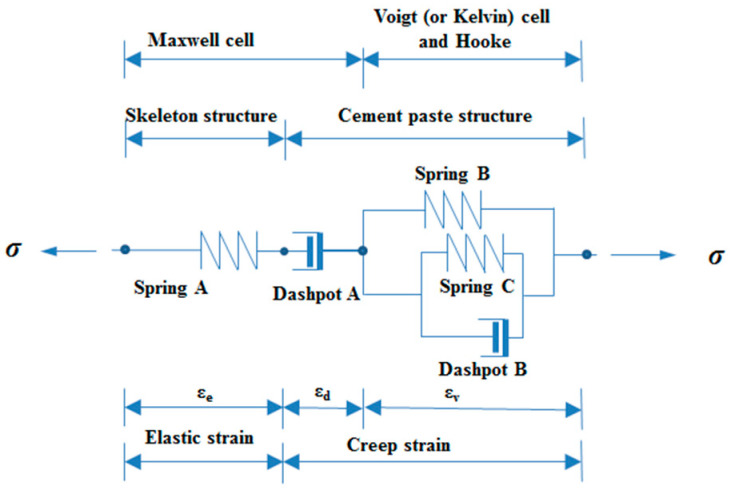
ZC rheological model of concrete creep.

**Figure 2 materials-16-01337-f002:**
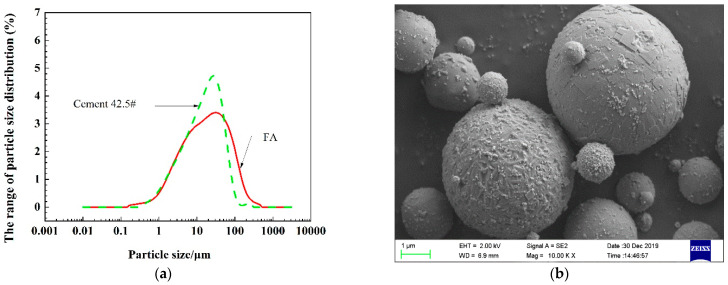
Powder particle size distribution and FA’s SEM image: (**a**) Powders’ particle size distribution; (**b**) FA’s SEM image.

**Figure 3 materials-16-01337-f003:**
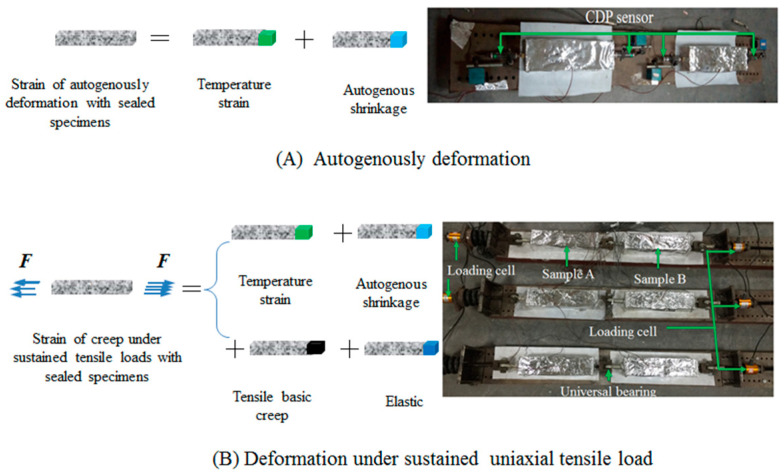
Experimental principle and physical picture of tensile creep strain divided.

**Figure 4 materials-16-01337-f004:**
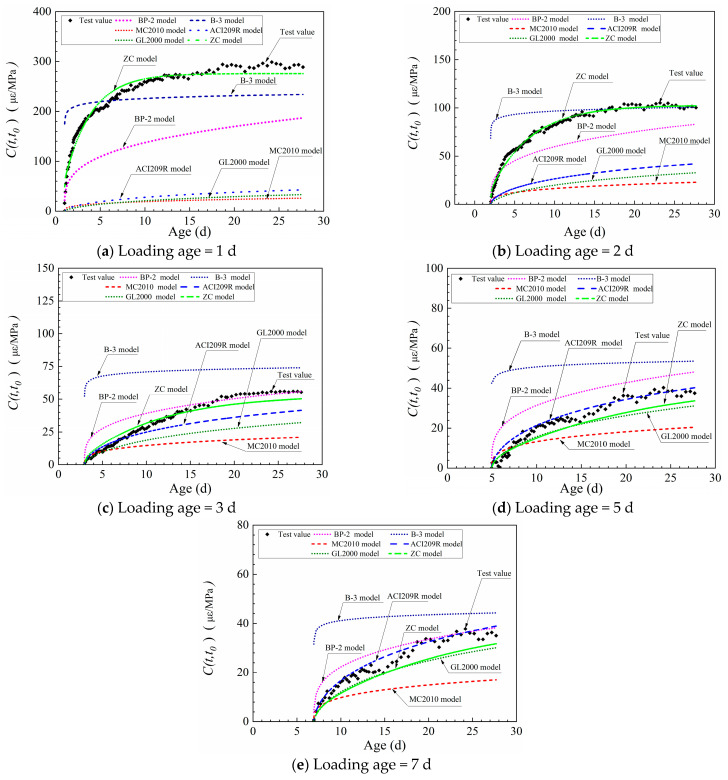
Comparison of specific creep models’ predicted values of FA30 with measured values.

**Figure 5 materials-16-01337-f005:**
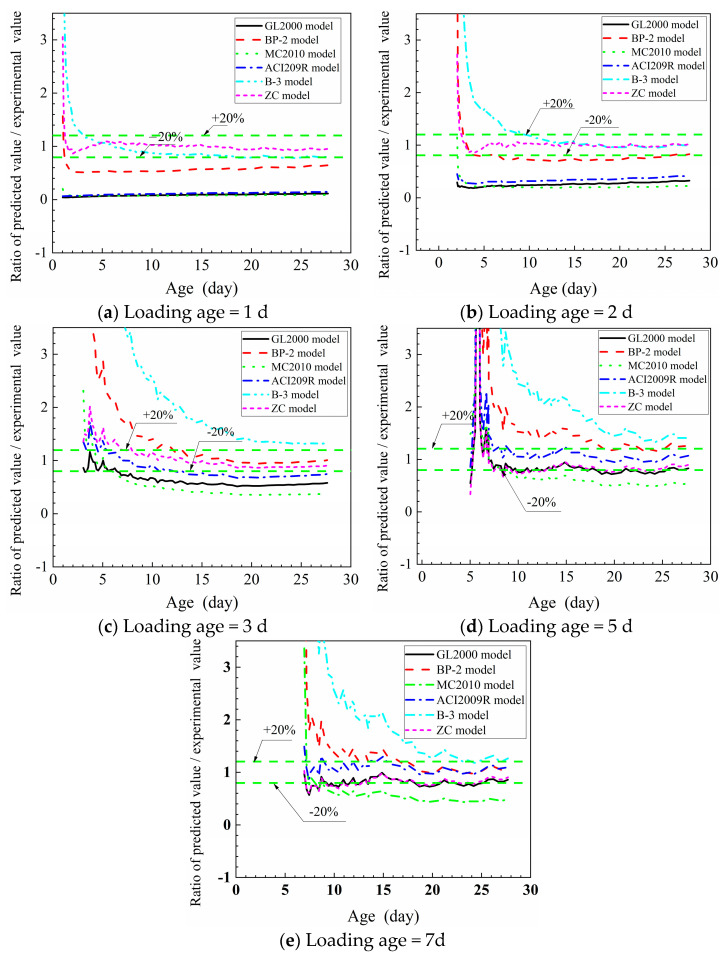
Precision analysis of prediction values of six models.

**Table 1 materials-16-01337-t001:** Factors being considered in six prediction models of tensile creep.

	Factors			Model			
		ZC	BP-2	B-3	MC2010	ACI209R	GL2000
Internal factors	Mass of aggregate	●		●			
	Air content	●				●	
	FA	●					
	Cement type	●	●	●	●	●	●
	The specific gravity of the fine aggregate	●				●	
	Slump					●	
	Water-binder ratio	●		●			
External factors	Loading age	●	●	●	●	●	●
	Calculating age	●	●	●	●	●	●
	Stress	●	●	●	●	●	●
	Cross-section shape						●
	28d strength	●	●	●	●	●	●
	Cross section size	●	●	●	●	●	●
	28d Elastic modulus	●			●	●	●
	Ambient humidity	●	●	●	●	●	●
	Ambient temperature				●		●
	Drying age						●

**Table 2 materials-16-01337-t002:** Properties of cement and FA.

Materials	Index											
	Chemical Composition (%)								Physical Properties			
	SiO_2_	Al_2_O_3_	Cao	Fe_2_O_3_	Na_2_O	MgO	SO_3_	K_2_O	Specific Surface Area (m^2^/cm^3^)	Density (g/cm^3^)	Ignition Loss (%)	Average Grain Size (μm)
Cement	24.95	6.99	54.33	2.83	0.21	2.16	2.89	0.66	368	2.99	3.66	26.04
FA	46.6	41.4	3.18	3.90	0.00	0.22	0.61	0.72	337	2.03	4.79	62.12

**Table 3 materials-16-01337-t003:** Mix Proportion of HSC with 30wt % FA (kg/m^3^).

Cement	Fly Ash	Sand	Coarse Aggregate	Superplasticizer	Water
391	167	687	988	11	167

**Table 4 materials-16-01337-t004:** Basic parameters of TC experiments and mechanical properties of HSC. (***W/B*** = 0.3, FA = 30wt %).

Loading Age (d)	Applied Load (kN)	Splitting Tensile Strength (MPa)	Compression Strength (Mpa)	Tensile Modulus of Elasticity (Gpa)
1	5.99	1.27	9.81	21.67
2	12.77	2.71	27.55	29.02
3	15.03	3.19	31.50	33.11
5	17.52	3.72	42.67	37.10
7	18.36	3.90	51.91	40.09

**Table 5 materials-16-01337-t005:** Calculated values of parameters in different prediction models.

**ZC model**	$\begin{array}{l} t_{0}\\ \left(d\right) \end{array}$	$\begin{array}{c} 1/\chi \varphi \\ \left(10^{-6}/MPa\right) \end{array}$	φ	$\begin{array}{c} 1/(E_{V}+E_{H})\\ \left(10^{-6}/MPa\right) \end{array}$		$E_{H}/E_{V}$	q
	1	214.65	0.33	61.36		22.74	2.76
	2	88.04	0.20	14.60		17.57	4.78
	3	47.03	0.11	6.20		10.86	6.52
	5	38.54	0.06	5.27		6.42	7.52
	7	36.24	0.06	5.87		5.98	7.90
**BP-2 model**	$\begin{array}{l} t_{0}\\ \left(d\right) \end{array}$	B	m	n		E′/GPa	
	1	1.49	0.29	0.28		21.31	
	2	1.49	0.29	0.28		39.13	
	3	1.49	0.29	0.28		50.58	
	5	1.49	0.29	0.28		50.81	
	7	1.49	0.29	0.28		56.88	
**B-3 model**	$\begin{array}{l} t_{0}\\ \left(d\right) \end{array}$	$r(t_{0})$	$Q\left(t_{0}\right)f$	$q_{2}$ of Modified B-3 model			
	1	9.70	0.77	566.75			
	2	9.85	0.57	332.44			
	3	9.94	0.48	292.18			
	5	10.06	0.39	266.37			
	7	10.15	0.33	256.65			
	w/c	a/c	$c(Kg/m^{3})$	$q_{2}$		$q_{3}$	$q_{4}$
	0.43	4.28	391.00	235.20		2.27	0.05
**MC2010 model**	$\begin{array}{l} t_{0}\\ \left(d\right) \end{array}$	$\beta (t_{0})$					
	1	0.91					
	2	0.80					
	3	0.74					
	5	0.68					
	7	0.64					
	$\beta (f_{cm})$	$\alpha_{1}$	$\alpha_{2}$	$\alpha_{3}$	$\varphi_{H}$	$\beta_{H}$	$E_{c}(28)/GPa$
	1.98	0.60	0.87	0.70	1.35	294.43	44.07
**ACI209R model**	S/(S + G)	*γ* * _φ_ *	Slump (mm)	*γ* * _s_ *			
	0.41	0.88	250	1.48			
**GL2000 model**	$t_{0}\left(d\right)$	v/s (mm)	*h*				
	As others	28.60	0.60				

## Data Availability

The data that support the findings of this study are available from the corresponding author upon reasonable request.

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
