# Peer review of "The Influence of Fly Ash on the Tensile Creep Prediction of High-Strength Concrete at Early Ages"

_materials, 2023, doi:10.3390/ma16041337_

Round 1
Reviewer 1 Report
An interesting manuscript on an important topic as cement replacement usage increases in an attempt to reduce portland cement consumption. As it currently stands, some of the potential impact is lost by the small but regular errors in the English, which should be relatively easy to put right with the appropriate help (a careful read through by a native English speaker with some experience in technical writing). I would also ask that you review the figures with a view to making them clearer and more easily interpreted. Finally, the conclusions would benefit from some expansion so as to include the significance of the results obtained with respect to the production of high strength/low creep concrete. I look forward to seeing the finalised manuscript.
Reviewer 2 Report
The results of “The influence of fly ash on the tensile creep prediction of high strength concrete at early ages” are of potential interest. The introduction section provides sufficient background of past literatures. In the experimental Programme section, all the testing methods are sufficiently described. In the experimental result and discussion section, the results are elaborately discussed with figures and tables. The conclusions are supported by the results. All the references are related to this research and also sufficient. However, the following corrections are to be carried out before the acceptance of the Manuscript.
1. Abstract: State the need of the study. Present your research recommendation. What is ZC?
2. Mention the novelty/research gap of your research.
3. Tables 2. Mention the unit of Specific surface area m2/cm3 in m2/kg. Also use squre value in superscript. Follow the same throughout the manuscript.
4. Many sub heading are there. But they are not numbered. Follow the journal template strictly.
5. Show more experimental photos.
6. Strengthen the conclusion section.
7. What is your recommendation/future scope of your research? Present it in the conclusion section.
8. Mention your research significance/impact in the manuscript.
Reviewer 3 Report
This manuscript evaluates the influence of fly ash on the tensile creep prediction of high strength concrete at early ages. The manuscript is elaborately described and contextualized with the help of previous and present theoretical background and empirical research. All the references cited are relevant to this area of research and also adequate. The methods are clearly stated. The result and discussion of the research are coherent and balanced. The conclusions are supported by the results. However, some minor corrections need to be addressed before the acceptance the Manuscript.
1. Abstract: Mention the research need. What is ZC model?
1. It would be better if key words are arranged in alphabetical order
2. Introduction needs to be strengthened.
3. What is the novelty of this research? Mention it.
4. Table 2. Use subscript properly in the chemical compounds. Use superscript for units. Follow the same throughout the manuscript.
5. Fig.3. Use clear image.
6. Where is the heading Results? Introduce it.
7. Compare your results with the past literatures.
8. Mention your research recommendation at the end of conclusion part.
